# ROYAL SOCIETY
# OPEN SCIENCE

biophysics/complexity

diabetes, pancreatic α-cell, glucagon, mitochondrial dysfunction, free fatty acid

**Authors for correspondence:**
Matjaž Perc
e-mail: matjaz.perc@um.si
Marko Marhl
e-mail: marko.marhl@um.si

†These authors contributed equally to this work.

# Modelling of dysregulated glucagon secretion in type 2 diabetes by considering mitochondrial alterations in pancreatic α-cells

Vladimir Grubelnik[1,†], Rene Markovič[1,2,†],
Saška Lipovšek[2,3,4,6], Gerd Leitinger[6], Marko Gosak[2,3],
Jurij Dolenšek[2,3], Ismael Valladolid-Acebes[7],
Per-Olof Berggren[7], Andraž Stožer[3], Matjaž Perc[2,8,9]
and Marko Marhl[2,3,5]

[1]Faculty of Electrical Engineering and Computer Science, [2]Faculty of Natural Sciences and Mathematics, [3]Faculty of Medicine, [4]Faculty of Chemistry and Chemical Engineering, and [5]Faculty of Education, University of Maribor, 2000 Maribor, Slovenia
[6]Division of Cell Biology, Histology and Embryology, Gottfried Schatz Research Center for Cell Signaling, Metabolism and Aging, Medical University of Graz, Neue Stiftingtalstrasse 6, 8010 Graz, Austria
[7]The Rolf Luft Research Center for Diabetes and Endocrinology, Karolinska Institutet, Karolinska University Hospital L1, 171 76 Stockholm, Sweden
[8]Department of Medical Research, China Medical University Hospital, China Medical University, Taichung, Taiwan
[9]Complexity Science Hub Vienna, 1080 Vienna, Austria

Type 2 diabetes mellitus (T2DM) has been associated with insulin resistance and the failure of β-cells to produce and secrete enough insulin as the disease progresses. However, clinical treatments based solely on insulin secretion and action have had limited success. The focus is therefore shifting towards α-cells, in particular to the dysregulated secretion of glucagon. Our qualitative electron-microscopy-based observations gave an indication that mitochondria in α-cells are altered in Western-diet-induced T2DM. In particular, α-cells extracted from mouse pancreatic tissue showed a lower density of mitochondria, a less expressed matrix and a lower number of cristae. These deformities in mitochondrial ultrastructure imply a decreased efficiency in mitochondrial ATP production, which prompted us to theoretically explore and clarify one of the most challenging problems associated with T2DM, namely the lack of glucagon secretion in hypoglycaemia and its oversecretion at high blood glucose concentrations. To this purpose, we

constructed a novel computational model that links α-cell metabolism with their electrical activity and glucagon secretion. Our results show that defective mitochondrial metabolism in α-cells can account for dysregulated glucagon secretion in T2DM, thus improving our understanding of T2DM pathophysiology and indicating possibilities for new clinical treatments.

## 1. Introduction

For several decades, diabetes research has been focusing on insulin resistance and the consequent defects in pancreatic β-cells and insulin secretion. Clinical therapies have evolved around this concept; however, with only limited success. Therefore, the role of pancreatic α-cells and glucagon secretion has been revisited and type 2 diabetes mellitus (T2DM) is considered as a bi-hormonal defect proposing that diabetic hyperglycaemia would not develop unless the lack of insulin was accompanied by hypersecretion of glucagon. Moreover, as Unger & Cherrington [1] have noted, glucagon excess, rather than insulin deficiency, is the *sine qua non* condition of diabetes. Glucagon secretion from α-cells most probably involves both intrinsic and paracrine mechanisms. Whether glucose inhibits α-cells directly or by paracrine mechanisms has been a matter of debate, and probably, the predominant level of control may depend on the physiological situation and species [2,3]. Moreover, it has been shown that glucose inhibits glucagon release at concentrations below the threshold for β-cell activation and insulin secretion, which would point more to intrinsic mechanisms of glucagon secretion in α-cells, at least in hypoglycaemic conditions [4]. Several concepts of this intrinsic glucagon secretion have been evolved, from store-operated models [5,6] to $K_{ATP}$-channel-centred models [7–9]; for a recent review of these α-cell-intrinsic models for glucagon secretion, see [2]. In this huge body of evidence supporting the intrinsic mechanisms of glucagon secretion in hypoglycaemic conditions, the $K_{ATP}$-channel-dependent glucose regulation of glucagon release is one of the most documented concepts [7–11]. The proposed mechanism is based on experimental results showing that glucose-induced inhibition of $K_{ATP}$ channels in α-cells results in inhibition of glucagon secretion [10]. The α-cell $K_{ATP}$-channel open probability is very low in low glucose, the net $K_{ATP}$-channel conductance at 1 mM glucose being around 50 pS, which is only around 1% of that in β-cells (3–9 nS) [10,12,13]. Therefore, in low glucose (1 mM), α-cells are electrically active and secrete glucagon. At higher glucose levels, the open probability of $K_{ATP}$ channels decreases even more, causing a further membrane depolarization, closing the voltage-dependent $Na^+$ channels, and decreasing the amplitude of action potential firing. This in turn reduces the amplitude of P/Q-type $Ca^{2+}$-currents and glucagon secretion [10].

In diabetes, secretion of glucagon is inadequately high at high glucose, exacerbating hyperglycaemia, and inadequately low at low glucose, possibly leading to fatal hypoglycaemia. Although the complete causal mechanisms remain unrevealed, there is experimental evidence showing that an increase in $K_{ATP}$-channel conductance mimics the glucagon secretory defects associated with T2DM. Treatment of non-diabetic mouse islets with oligomycin [10] and dinitrophenol [14], which inhibit mitochondrial ATP synthase and thus increase the $K_{ATP}$-channel conductance, cause typical T2DM 'right-shift' in glucagon secretion, i.e. inadequate secretion at low glucose and unsuppressed secretion at high glucose. Conversely, the $K_{ATP}$-channel blocker tolbutamide is at least partly able to restore glucose inhibition of glucagon secretion in T2DM islets [10,11]. In summary, these data indicate that metabolism importantly controls glucagon secretion. α-Cells need sufficient ATP supply, in particular an efficient mitochondrial function to maintain glucagon secretion at low glucose, and effective glycolysis as a switch for glucose-induced inhibition of glucagon secretion. The oxidative metabolism in mitochondria needs to produce enough ATP to keep $K_{ATP}$-channel conductance low and ensure a fine-regulated glucagon secretion [10]. This indicates that impaired mitochondrial structure and function in α-cells could be one of the main culprits for the dysregulated glucagon secretion.

In pancreatic tissue, mitochondrial dysfunction was established as one of the major causes for impaired secretory response of β-cells to glucose [15,16]. Also, it has been proposed that functional and molecular alterations of β-cells, rather than a decrease in β-cell mass, account for insufficient β-cell functional mass in T2DM [17–19]. In T2DM, β-cells contain swollen mitochondria with disordered cristae [20–22] and display an impaired stimulus-secretion coupling. An insufficient insulin secretion is also linked with a reduced hyperpolarization of mitochondrial inner-membrane potential, partially via increased UCP-2 expression, and a reduced glucose-stimulated ATP/ADP ratio [20,21]. In good agreement with the above, it has been shown that mitochondrial oxidative phosphorylation decreases by 30–40% in insulin-resistant subjects [23,24].

Pancreatic α-cells are also affected in obesity and T2DM. Experimental studies have shown that α-cell area is reduced in obese mice as a result of cell hypotrophy, and that an increased apoptosis and decreased proliferation are present [25]. The morphology of α-cells has also been studied with electron microscopy [26,27]. However, to the best of our knowledge, there have been no systematic studies of changes in α-cell ultrastructure in T2DM. Moreover, it is inherently challenging to functionally study α-cell mitochondrial metabolism. They namely represent a minor proportion of islet cells, are located at the periphery of the mouse islets, where they are subjected to stress during isolation, and they have been reported to gradually disappear during culture [28,29]. In the present study, we address this issue and reveal that the morphology of mitochondria in α-cells of Western-diet-induced diabetic mice is changed considerably, which implies a less efficient metabolism in α-cells of mice with T2DM. This observation directed us into constructing a mathematical model of the α-cell that incorporates cell metabolism and energetics, electrical activity and glucagon secretion. Our theoretical findings suggest that the attenuated mitochondrial oxidative capacity could explain the dysregulation of glucagon secretion that is typical for T2DM.

# 2. Material and methods

## 2.1. Animals, diets and metabolic parameters

Animal experiments were performed in 12-week-old male C57BL6/J mice purchased from Charles River. Animals were housed three per cage in a constant 12 : 12 light/day cycle and with ad libitum access to food and water. At start, 12 mice with equivalent body weight were randomly divided into two groups of 6 mice each and given either the Western diet (WD) (4.7 kcal g$^{-1}$, % kcal: 43 carbohydrates, 40 fat, 17 proteins, Research Diets Inc.) or the regular chow diet (3.0 kcal g$^{-1}$, % kcal: 71.7 carbohydrates, 10.5 fat, 17.7 proteins, R70, Lantmännen) ad libitum for eight weeks. At the end of the diet intervention, body weight was determined, blood glucose was measured using a FreeStyle Glucometer (Abbot Diabetes Care), and blood samples were collected from the tail vein into Microvette tubes (Sarsted) for serum analyses. Thereafter, mice were euthanized by cervical dislocation, and their abdomens exposed to isolate the pancreatic tissue for electron microscopy. Serum insulin levels were determined using an ultrasensitive mouse insulin ELISA kit (Crystal Chem Inc.). All animal studies were done in accordance with the guidelines from local authorities and ethical committees, i.e. the Stockholm Northern Animal Experiments Ethics Board, and in accordance with the Directive 2010/63/EU of the European Parliament and of the Council on the Protection of Animals Used for Scientific Purposes.

## 2.2. Tissue preparation for transmission electron microscopy

Small pieces of pancreas were fixed in 2.45% glutaraldehyde and 2.45% paraformaldehyde in a 0.1 M sodium cacodylate buffer (pH 7.4) at room temperature for 3 h and at 4°C for 14 h, washed in a 0.1 M sodium cacodylate buffer (pH 7.4) at room temperature for 3 h and postfixed with 2% $OsO_4$ at room temperature for 2 h. The tissue was dehydrated in a graded series of ethanol (50, 70, 90, 96, 100%, each for 30 min at room temperature) and embedded in TAAB epoxy resin (Agar Scientific Ltd, Essex, UK). For transmission electron microscopy (TEM), ultrathin sections (75 nm) were transferred onto copper grids, stained with uranyl acetate and lead citrate and analysed by a Zeiss EM 902 transmission electron microscope. Pancreatic tissue slices that contained islets of Langerhans from normal and WD-fed mice were examined.

The tissue of six mice fed with a WD for eight weeks and the tissue of six mice fed with the regular chow have been examined. From the pancreas of each individual, we analysed six different pieces of tissue containing islets of Langerhans. The person evaluating the electron microscopy slices was blinded to the treatment group.

## 2.3. Mathematical α-cell model

We conducted a computational α-cell model to simulate and explore several interconnected steps in the glucose-dependent signalling cascade from the initial metabolic processes to exocytosis. We combined a mathematical model for glycolysis and glucose-driven mitochondrial activity [30] with a model for simulating α-cell electrical and $Ca^{2+}$ activity and finally with glucagon secretion [31]. This unique coupling of the α-cell metabolism with the electrical activity enabled us to study the interplay

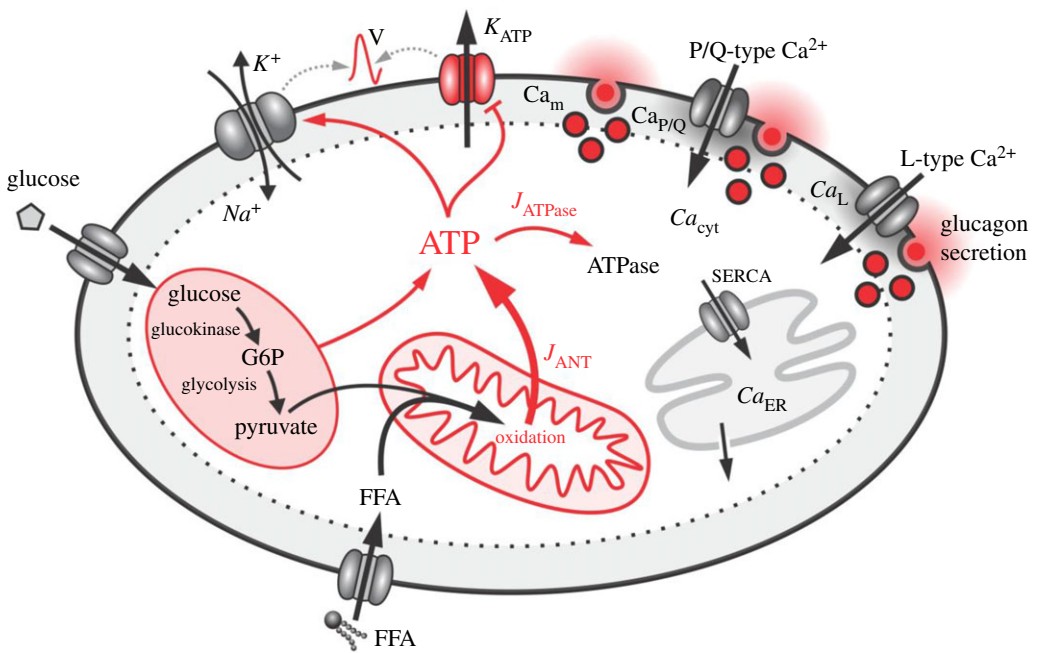

**Figure 1.** Overview of the main mechanisms of the holistic α-cell computational model. Glycolysis and mitochondria produce ATP which reduces $K_{ATP}$-channel conductance and enhances $Na^+/K^+$ ATPases and in turn determines α-cell electrical activity. The latter regulates exocytosis via voltage-gated P/Q- and L-type $Ca^{2+}$ channels. For further explanations, see text.

between processes related to glucagon secretion and ATP production in mitochondria. Most importantly, we adjusted the model and the parameters in order to fit several aspects of model predictions with experimental findings. In addition, we interconnected glucose and free fatty acid (FFA) metabolism with a mechanism that enables the α-cell to regulate its energetics in a stimulation-dependent manner. We have also included an interaction between cytosolic ATP concentration and the activity of ATPases. Figure 1 features the scheme of the computational model with highlighted crucial processes that are involved in ATP production and regulation of glucagon secretion. The whole mathematical model was written in Berkeley Madonna software (University of California at Berkeley, California, USA) and C++. A detailed description of the individual model components is given in the continuation of the text, whereas the sensitivity analysis of the model is presented in the electronic supplementary material.

### 2.3.1. Glucose uptake and glycolysis

The first step in glucose metabolism is glucose uptake. An accurate glucose sensing is linked with a rapid transport of glucose from the interstitium to the cytosol. A fast transport of glucose avoids a delay in equilibration between the extra- and intracellular glucose concentrations and prevents the drop in free cytosolic glucose concentration due to phosphorylation [32]. A rapid glucose transport is a hallmark of β-cells, since a good sensing is essential for their function. In β-cells, the rapid glucose transport is principally ensured by the efficient glucose transporter Glut2. α-Cells, by contrast, do not express Glut2, but Glut1. The mathematical formalism for describing glucose transport into the intracellular space is based on the β-cells model proposed by Pedersen *et al.* [33]

$$J_{GK} = \frac{V_{max,GK}G}{K_{m,GK} + G},$$ (2.1)

where $J_{GK}$ is the glucokinase reaction rate, $V_{max,GK}$ is the maximum reaction rate, $K_{m,GK}$ is the half-saturation constant and $G$ is the stimulatory glucose concentration. Experiments with D-glucose and its non-metabolizable analogue, 3-O-MG, showed that the glucose uptake is much slower in α-cells compared with that in β-cells [34,35]. We adjusted the values of the maximum reaction rate $V_{max,GK}$ and half-saturation constant $K_{m,GK}$, so that they reflect experimentally determined differences in glucose uptake between α- and β-cells. In particular, characteristic values of the 3-O-MG transport in α-cells were $K_{m,GK} = 8.5$ mM and $V_{max,GK} = 1\ 10^{-3}$ µM ms$^{-1}$ [34,35].

In continuation, we link $J_{GK}$ to the glycolytic part of the model that is based on the theoretical framework proposed by Smolen [30]. The aim of the model was to simulate the kinetics of skeletal

**Table 1.** Parameter values for the glucose uptake and glycolysis model.

| $V_{max,GK} = 1 \times 10^{-3}$ μM ms$^{-1}$ | $K_{m,GK} = 8.5$ mM | | | |
|---|---|---|---|---|
| $k_{GPDH} = 0.0005$ μM ms$^{-1}$ | $V_{max,PFK} = 0.05$ μM ms$^{-1}$ | $\lambda = 0.06$ | | |
| $K_1 = 30$ μM | $K_2 = 1$ μM | $K_3 = 50\,000$ μM | $K_4 = 220$ μM | |
| $f_{13} = 0.02$ | $f_{23} = 0.2$ | $f_{41} = 20$ | $f_{42} = 20$ | $f_{43} = 20$ |

muscle phosphofructokinase (PFK) as a function of AMP, ATP, fructose 6-phosphate concentration (F6P), glucose 6-phosphate (G6P), and fructose 1,6-bisphosphate concentration (FBP). An important feature of the model is the activity level of the enzyme glucokinase (GK), a glucose sensor, which provides the input for the enzyme PFK. The main components of the mathematical model are defined as

$$\frac{dG6P}{dt} = J_{GK} - J_{PFK},$$ (2.2)

$$\frac{dFBP}{dt} = J_{PFK} - \frac{1}{2}J_{GPDH},$$ (2.3)

$$J_{PFK} = V_{max,PFK} \frac{(1 - \lambda)w_{1110} + \lambda \sum_{abc} w_{abc1}}{\sum_{abcd} w_{abcd}}$$ (2.4)

and

$$J_{GPDH} = k_{GPDH}\sqrt{\frac{FBP}{1\,\mu M}}\mu M \; ms^{-1}.$$ (2.5)

where $J_{GPDH}$ stands for the glyceraldehyde 3-P dehydrogenase (GPDH) reaction rate and $J_{PFK}$ is the PFK reaction rate. G6P is assumed to be in rapid equilibrium with F6P (F6P = 0.3 G6P). $J_{GK}$ is the glucokinase reaction rate and is a glucose-dependent parameter (see equation (2.1)). The selected values for $J_{GK}$ lead to glycolytic oscillations as proposed by Smolen [30]. Additionally, the values of $J_{GPDH}$ are comparable with experimentally measured values at 1 and 10 mM glucose [34,36]. Lastly, the parameter $w_{abcd}$ in the PFK reaction rate is given by

$$w_{abcd} = \frac{1}{f_{13}^{ab}f_{23}^{bc}f_{41}^{ad}f_{42}^{bd}f_{43}^{cd}}\left(\frac{AMP}{K_1}\right)^a\left(\frac{FBP}{K_2}\right)^b\left(\frac{F6P}{K_3}\right)^c\left(\frac{ATP}{K_4}\right)^d,$$ (2.6)

where $w_{abcd}$ reassembles the fraction of PFK in state *abcd*, whereby *a*, *b*, *c* and *d* are either 1 or 0, as described previously [30,33,37]. Parameter values for the glucose uptake and glycolytic part of the model described in equations (2.1)–(2.6) are given in table 1.

### 2.3.2. Glucose and free fatty acid oxidation

Many tissues are using a variety of carbon-based energy sources to maintain ATP production, predominately with β-oxidation of FFAs [3]. The latter has been shown to regulate glucose-induced insulin secretion in pancreatic islets [38]. Much less is known about the role of FFAs in regulating glucagon secretion. However, it has been shown that short-term exposure to supra-physiological levels of FFAs increases glucagon secretion [39]. In particular at low glucose levels, when glucagon is secreted in larger amounts, β-oxidation of FFAs can provide substantial amounts of ATP for the processes regulating the glucagon secretion in α-cells. Under hypoglycaemic conditions, it has indeed been shown that FFAs are pivotal and contribute to ATP production, which maintains glucagon secretion by energizing the Na$^+$/K$^+$ pump [3]. We have developed a computational model that takes into account both energy sources, glucose and FFA, for mitochondrial ATP production as follows:

$$J_{GO} = k_{GO}(1 - k_{md})J_{GPDH}$$ (2.7)

and

$$J_{FFAO} = k_{FFAO}(1 - k_R G)(1 - k_{md}).$$ (2.8)

In the above equations, $J_{GO}$ is the glucose oxidation rate, $k_{GO}$ is the net yield of ATPs per glucose, β-oxidation of FFAs is given $J_{FFAO}$, $k_{FFAO}$ is the β-oxidation rate constant, $k_R$ is the glucose reduction factor representing the effect of the Randle cycle [40] and $k_{md}$ is the level of mitochondrial dysfunction

**Table 2.** Parameter values for the mitochondrial ATP production part of the mathematical model.

| | | |
|---|---|---|
| $k_{G0} = 38/2$ | $k_R = 0.01\ \text{mM}^{-1}$ | $k_{FFA0} = 0.1\ \mu\text{M ms}^{-1}$ |
| $k_{ATPase} = 5 \times 10^{-5}\ \text{ms}^{-1}$ | $k_{ATPase,r} = 0.7$ | $A_{tot} = 2700\ \mu\text{M}$ |

(i.e. $k_{md} = 0.3$ reflects a 30% decrease in mitochondrial function). The dynamics of cytosolic ATP concentration is modelled as

$$J_{ANT} = J_{GO} + J_{FFAO}, \tag{2.9}$$

$$J_{ATPase} = k_{ATPase}(1 - k_{ATPase,r}k_{md})\text{ATP}, \tag{2.10}$$

$$\frac{\text{dATP}}{\text{d}t} = -J_{GK} - J_{PFK} - J_{ATPase} + 2J_{GPDH} + J_{ANT} \tag{2.11}$$

and

$$A_{tot} = \text{ATP} + \text{ADP}. \tag{2.12}$$

The mitochondria contribute to cytosolic ATP with the adenine nucleotide translocator ($J_{ANT}$), which exchanges free ATP with free ADP across the inner mitochondrial membrane. Overall, glycolysis ($J_{GPDH}$) also contributes to ATP. Major consumers of ATP described in equation (2.11) are the glucokinase ($J_{GK}$), PFK ($J_{PFK}$) and ATPases ($J_{ATPase}$). Equation (2.10) mimics the reduction in ATP production due to mitochondrial dysfunction ($k_{md}$) and the corresponding ATPases reduction $k_{ATPase,r}$. The cytosolic ADP concentration is acquired from the conservation law (equation (2.12)). The parameter values for this segment of the model are given in table 2.

This mathematical model described with equations (2.7)–(2.12) enabled us to simulate several α-cell-specific features regarding glucose and FFA oxidation. In α-cells, it has been shown that the glucose uptake mechanism probably represents a rate-limiting step in glucose metabolism [34]. Much more glucose is oxidized in β-cells where the dose–response curve of glucose oxidation displays a sigmoidal shape with an approximately sixfold larger saturated glucose oxidation rate in comparison with α-cells [34]. Noteworthy, β-cells metabolize glucose essentially via aerobic glycolysis, whereas the glycolysis in α-cells is largely anaerobic [41,42]. Consequently, the lower coupling between glycolysis and ATP synthesis in mitochondria explains the fact that in α-cells only slight increases in relative cytosolic ATP [43,44] and a nearly constant ATP/ADP ratio [36,45] can be observed.

### 2.3.3. Glucagon secretion

The model for simulating α-cell electrical activity and exocytosis is a modified version of previous endeavours [31,46]. The aim of this part of the model is to link metabolic processes and glucagon secretion. The α-cell electrical activity is defined as

$$\frac{\text{d}V}{\text{d}t} = -\frac{I_{CaL} + I_{CaP/Q} + I_{CaT} + (I_{Na} + I_K + I_{KA})p_{ATP} + I_{KATP} + I_L + I_{SOC}}{C}, \tag{2.13}$$

where $V$ and $C$ are the membrane voltage and capacitance of the α-cell. $I_{CaL}$, $I_{CaP/Q}$ and $I_{CaT}$ are voltage-dependent $Ca^{2+}$ currents, $I_{Na}$ is a voltage-dependent $Na^+$ current, $I_K$ is a delayed rectifier $K^+$ current, $I_{KA}$ is an A-type voltage-dependent $K^+$ current, $I_L$ is a leak current, $I_{SOC}$ is a store-operated $Ca^{2+}$ current (SOC) and $I_{KATP}$ is an ATP-sensitive $K^+$ current which we link to the ratio between cytosolic ATP and ADP concentration. The ratio between cytosolic ATP and ADP is defined as

$$\text{RAT} = \frac{\text{ATP}}{\text{ADP}}. \tag{2.14}$$

The relation between RAT and $K_{ATP}$-channel conductance is driven by a complex series of signalling pathways [47,48]. To mimic the resulting transduction delays and smoothing of the activity profiles by the signalling cascade [49], we compute the smoothed $RAT_f$ signal as

$$\frac{\text{dRAT}_f}{\text{d}t} = k_f(\text{RAT} - \text{RAT}_f), \tag{2.15}$$

where $k_f$ regulates the level of smoothing. The smoothed signal $RAT_f$ is then used for the calculation of the $K_{ATP}$-channel conductance, with the following equation

$$g_{K,ATP}(\text{RAT}_f) = g_1 e^{-k_{g,KATP}\text{RAT}_f}. \tag{2.16}$$

The parameters $g_1$ and $k_{g,KATP}$ in equation (2.16) were determined by fitting an exponential function to the experimentally measured values of $g_{K,ATP}$ conductance and the average ATP/ADP ratio at 1 and 6 mM glucose [50]. In this manner, the cytosolic ATP concentration is directly linked to the conductance of $K_{ATP}$ channels. Lastly, the additional parameter $p_{ATP}$ in equation (2.13) mimics the sodium–potassium exchanger dependence on ATP. Namely, reduction in ATP concentration affects the activity of this exchanger. It has been shown that the effect is more profound for very low ATP concentrations that correspond to less than 1 mM glucose [3]. In the model, this aspect was incorporated by defining $p_{ATP}$ as

$$p_{\mathrm{ATP}} = \frac{1}{1 + (g_{\mathrm{K,ATP}}(\mathrm{RAT}_f)/0.35 \text{ nS})^8}. \tag{2.17}$$

The individual currents in (equation (2.13)) are defined by the following equations

$$I_{\mathrm{CaL}} = g_{\mathrm{CaL}} m_{\mathrm{CaL}}^2 h_{\mathrm{CaL}} (V - V_{\mathrm{Ca}}), \tag{2.18}$$

$$I_{\mathrm{CaP/Q}} = g_{\mathrm{CaP/Q}} m_{\mathrm{CaP/Q}} h_{\mathrm{CaP/Q}} (V - V_{\mathrm{Ca}}), \tag{2.19}$$

$$I_{\mathrm{CaT}} = g_{\mathrm{CaT}} m_{\mathrm{CaT}}^3 h_{\mathrm{CaT}} (V - V_{\mathrm{Ca}}), \tag{2.20}$$

$$I_{\mathrm{Na}} = g_{\mathrm{Na}} m_{\mathrm{Na}}^3 h_{\mathrm{Na}} (V - V_{\mathrm{Na}}), \tag{2.21}$$

$$I_{\mathrm{K}} = g_{\mathrm{K}} m_{\mathrm{K}}^4 h_{\mathrm{K}} (V - V_{\mathrm{K}}), \tag{2.22}$$

$$I_{\mathrm{KATP}} = g_{\mathrm{K,ATP}}(\mathrm{RAT}_f) m_{\mathrm{KATP}}^2 h_{\mathrm{KATP}} (V - V_{\mathrm{K}}), \tag{2.23}$$

$$I_{\mathrm{KA}} = g_{\mathrm{KA}} m_{\mathrm{KA}}^2 h_{\mathrm{KA}} (V - V_{\mathrm{K}}), \tag{2.24}$$

$$I_{\mathrm{L}} = g_{\mathrm{L}} (V - V_{\mathrm{L}}) \tag{2.25}$$

and

$$I_{\mathrm{SOC}} = g_{\mathrm{SOC}} (V - V_{\mathrm{SOC}}). \tag{2.26}$$

In equations (2.18)–(2.26), $g_x$ and $V_x$ represent the conductance and reverse potential of channels, respectively, and $x$ stands for the specific channel type. The activation and inactivation variables of channel $x$ are given by $m_x$ and $h_x$ and are defined as

$$\frac{\mathrm{d}m_x}{\mathrm{d}t} = \frac{m_{x,\infty}(V) - m_x}{\tau_{mx}(V)} \tag{2.27}$$

and

$$\frac{\mathrm{d}h_x}{\mathrm{d}t} = \frac{h_{x,\infty}(V) - h_x}{\tau_{hx}(V)}, \tag{2.28}$$

where $\tau_{mx}(V)$ and $\tau_{hx}(V)$ are time constants for $m_x$ and $h_x$. The steady-state activation and inactivation curves, $m_{x,\infty}(V)$ and $h_{x,\infty}(V)$, follow a Boltzmann function

$$m_{x,\infty}(V) = \frac{1}{1 + \mathrm{e}^{-(V - V_{mx}/S_{mx})}} \tag{2.29}$$

and

$$h_{x,\infty}(V) = \frac{1}{1 + \mathrm{e}^{-(V - V_{hx}/S_{hx})}}, \tag{2.30}$$

whereas the time constants are bell-shaped functions

$$\tau_{mx}(V) = \frac{\tau_{mVx}}{\mathrm{e}^{-((V - V_{\tau mx})/S_{\tau mx})} + \mathrm{e}^{((V - V_{\tau mx})/S_{\tau mx})}} + \tau_{m0x} \tag{2.31}$$

and

$$\tau_{hx}(V) = \frac{\tau_{hVx}}{\mathrm{e}^{-((V - V_{\tau hx})/S_{\tau hx})} + \mathrm{e}^{((V - V_{\tau hx})/S_{\tau hx})}} + \tau_{h0x}. \tag{2.32}$$

This electrical activity triggers Ca$^{2+}$ signals and glucagon secretion. Ca$^{2+}$ dynamics is modelled as

$$Ca_{P/Q_0} = Ca_m - \alpha \frac{i_{CaP/Q}}{B_{\mu d}Vol_{\mu d}}, \tag{2.33}$$

$$Ca_{L_0} = Ca_m - \alpha \frac{i_{CaL}}{B_{\mu d}Vol_{\mu d}}, \tag{2.34}$$

$$i_{CaP/Q} = g_{CaP/Q} \frac{(V - V_{Ca})}{N_{P/Q}}, \tag{2.35}$$

$$i_{CaL} = g_{CaL} \frac{(V - V_{Ca})}{N_L}, \tag{2.36}$$

$$Ca_{P/Q_C} = Ca_m, \tag{2.37}$$

$$Ca_{L_C} = Ca_m, \tag{2.38}$$

$$\frac{dCa_m}{dt} = -f \frac{\alpha I_{CaT}}{Vol_m} + f \frac{N_{P/Q}Vol_{\mu d}}{Vol_m} B_{\mu d} m_{CaP/Q} h_{CaP/Q}(Ca_{P/Q_0} - Ca_m)$$
$$+ f \frac{N_L Vol_{\mu d}}{Vol_m} B_{\mu d} m_{CaL} h_{CaL}(Ca_{L_0} - Ca_m) - f \frac{Vol_c}{Vol_m} k_{PMCA}Ca_m - f \frac{Vol_c}{Vol_m} B_m(Ca_m - Ca_c), \tag{2.39}$$

$$\frac{dCa_c}{dt} = -f(B_m(Ca_m - Ca_c) + p_{leak}(Ca_{er} - Ca_c) - k_{serca}Ca_{er}) \tag{2.40}$$

and

$$\frac{dCa_{er}}{dt} = -f \frac{Vol_c}{Vol_{er}}(p_{leak}(Ca_{er} - Ca_c) - k_{serca}Ca_{er}). \tag{2.41}$$

Equations (2.33)–(2.41) describe the dynamics of five main Ca$^{2+}$ domains: cytosol Ca$_c$, endoplasmic reticulum Ca$_{er}$, submembrane (Ca$_m$) and the microdomain Ca$^{+2}$ concentrations surrounding P/Q-type (Ca$_{P/Q_0}$, Ca$_{P/Q_c}$) and L-type (Ca$_{L_0}$, Ca$_{L_c}$) channels. In α-cells, Ca$^{2+}$ predominately enters through L-type channels, but exocytosis is mediated by P/Q-type channels [51]. Finally, glucagon secretion in the P/Q-type and L-type microdomains is defined as follows:

$$GS_{P/Q}(t) = m_{CaP/Q}h_{CaP/Q}f_H(Ca_{P/Q_0}, K_{P/Q}, n_{P/Q}) + (1 - m_{CaP/Q}h_{CaP/Q})f_H(Ca_{P/Q_c}, K_{P/Q}, n_{P/Q}), \tag{2.42}$$

$$GS_L(t) = m_{CaL}h_{CaL}f_H(Ca_{L_0}, K_L, n_L) + (1 - m_{CaL}h_{CaL})f_H(Ca_{L_c}, K_L, n_L), \tag{2.43}$$

$$GS_m(t) = f_H(Ca_m, K_m, n_m) \tag{2.44}$$

and

$$f_H(x, K, n) = \frac{x^n}{x^n + K^n}. \tag{2.45}$$

As described in more detail in [31]. The overall secreted glucagon is finally computed as

$$GS = \int_0^T (GS_{P/Q}(t) + GS_L(t) + GS_m(t)) \, dt. \tag{2.46}$$

Table 3 features the values of all parameters for equations (2.13)–(2.46).

## 3. Results

The systemic tests on WD-fed mice showed that after eight weeks, male C57BL6/J mice become obese and develop partially decompensated T2DM with hyperglycaemia and hyperinsulinaemia (see electronic supplementary material, table S1), which is in agreement with previously reported data [52]. The examination of pancreatic tissue by electron microscopy showed diet-induced alterations of mitochondrial morphology in α-cells. These morphological changes imply possible alterations in their function; however, at this stage, we are lacking experimental evidence about the mitochondrial function. Therefore, a link between the observed changes in the morphology and the corresponding functional alterations in mitochondria and α-cell secretory function has been investigated by means of mathematical modelling. The model predictions show that impaired bioenergetics with less efficient ATP production in α-cells could explain the 'right-shift' of glucagon secretion to higher glucose concentrations, as one of the usually observed hallmarks in T2DM. First, the results of the electron microscopy are presented, and in the continuation, the results of the mathematical model.

**Table 3.** Parameter values for the membrane potential, $Ca^{2+}$ dynamics and glucagon secretion model.

| | | | | | |
|---|---|---|---|---|---|
| $V_{Ca}$ | 65 mV | $\tau_{mVK}$ | 1.5 ms | $S_{mKa}$ | 10 mV |
| $V_K$ | −75 mV | $\tau_{m0K}$ | 15 ms | $V_{hKa}$ | −68 mV |
| $V_{SOC}$ | 65 mV | $V_{\tau mK}$ | −10 mV | $S_{hKa}$ | −10 mV |
| $g_{CaL}$ | 0.85 nS | $S_{\tau mK}$ | 25 mV | $\tau_{mVKa}$ | 0 ms |
| $g_{CaT}$ | 0.4 nS | $S_{mCaT}$ | 4 mV | $\tau_{m0Ka}$ | 0.1 ms |
| $V_{Na}$ | 70 mV | $V_{hCaT}$ | −52 mV | $\tau_{hVKa}$ | 60 ms |
| $V_L$ | −26 mV | $S_{hCaT}$ | −5 mV | $\tau_{h0Ka}$ | 5 ms |
| $g_{SOC}$ | 0.028 nS | $\tau_{mVCaT}$ | 15 ms | $V_{\tau hKa}$ | 5 mV |
| $g_{CaP/Q}$ | 0.35 nS | $\tau_{m0CaT}$ | 0 ms | $S_{\tau hKa}$ | 20 mV |
| $g_{Na}$ | 11 nS | $V_{\tau mCaT}$ | −50 mV | $C_m$ | 5 pF |
| $g_K$ | 4.5 nS | $S_{\tau mCaT}$ | 12 mV | $k_f$ | 0.01 s$^{-1}$ |
| $g_{Ka}$ | 1 nS | $\tau_{hVCaT}$ | 20 ms | $g_1$ | 0.5 nS |
| $g_L$ | 0.2 nS | $\tau_{h0CaT}$ | 5 ms | $k_{g,katp}$ | 0.2 |
| $g_{KATP}$ | 0.3 nS | $V_{\tau hCaT}$ | −50 mV | $f$ | 0.01 |
| $V_{mCaL}$ | −30 mV | $S_{\tau hCaT}$ | 15 mV | $V_{ol\mu d}$ | $2.618 \times 10^{-19}$ l |
| $S_{mCaL}$ | 10 mV | $V_{mNa}$ | −30 mV | $V_{olc}$ | $5.725 \times 10^{-13}$ l |
| $V_{mCaP/Q}$ | −1 mV | $S_{mNa}$ | 4 mV | $V_{olc}/V_{oler}$ | 31 |
| $S_{mCaP/Q}$ | 4 mV | $V_{hNa}$ | −52 mV | $N_{P/Q}$ | 100 |
| $V_{hCaL}$, $V_{hCaP/Q}$ | −33 mV | $S_{hNa}$ | −8 mV | $N_L$ | 400 |
| $S_{hCaL}$, $S_{hCaP/Q}$ | −5 mV | $\tau_{mVNa}$ | 6 ms | $k_{PMCA}$ | 0.3 ms$^{-1}$ |
| $\tau_{mVCaL}$, $\tau_{mVCaP/Q}$ | 1 ms | $\tau_{m0Na}$ | 0.05 ms | $k_{SERCA}$ | 0.1 ms$^{-1}$ |
| $\tau_{m0CaL}$, $\tau_{m0CaP/Q}$ | 0.05 ms | $V_{\tau mNa}$ | −50 mV | $p_{leak}$ | $3 \times 10^{-4}$ ms$^{-1}$ |
| $V_{\tau mCaL}$, $V_{\tau hCaL}$ | −23 mV | $S_{\tau mNa}$ | 10 mV | $n_{P/Q}$ | 4 |
| $S_{\tau mCaL}$, $S_{\tau hCaL}$ | 20 mV | $\tau_{hVNa}$ | 120 ms | $K_{P/Q}$ | 2 μM |
| $\tau_{hVCaL}$, $\tau_{hVCaP/Q}$ | 60 ms | $\tau_{h0Na}$ | 0.5 ms | $n_L$ | 4 |
| $\tau_{h0CaL}$, $\tau_{h0CaP/Q}$ | 51 ms | $V_{\tau hNa}$ | −50 mV | $K_m$ | 2 μM |
| $V_{mK}$ | −25 mV | $S_{\tau hNa}$ | 8 mV | $\alpha$ | $5.18 \times 10^{-15}$ μmol pA$^{-1}$ ms$^{-1}$ |
| $S_{mK}$ | 23 mV | $V_{mKa}$ | −45 mV | $V_{olm}$ | $5.149 \times 10^{-14}$ l |

## 3.1. Electron microscopy

The general structure of α-cells in WD-fed mice was comparable to that of control mice. The significant difference was in the structure of the mitochondria. In the control sample, the mitochondria were elongated and the inner and outer membrane showed a typical structure (figure 2a). In α-cells from WD-fed mice, the mitochondria were oval in shape, and the membranes were changed, the matrix was less well expressed and the number of cristae was decreased (figure 2b), which points towards a less efficient metabolism in these α-cells. These findings indicate that the morphology of mitochondria in α-cells is considerably altered in diabetic conditions. Additional electron microscopy images taken from all animals subjected to WD and from the control group are presented in the electronic supplementary material, figures S1–S12.

## 3.2. Theoretical insights

We have developed a comprehensive computational model that links the metabolic processes with the known electrophysiological and exocytotic properties of α-cells. The model incorporates the glucose uptake, glycolysis, glucose and FFA oxidation, mitochondrial ATP production, electrical activity, compartmentalized $Ca^{2+}$ dynamics and glucagon secretion mechanisms. The model is designed to simulate the ATP production and the ATP-related glucagon secretion in α-cells. A more detailed description of the processes and molecular mechanisms is given in Material and methods.

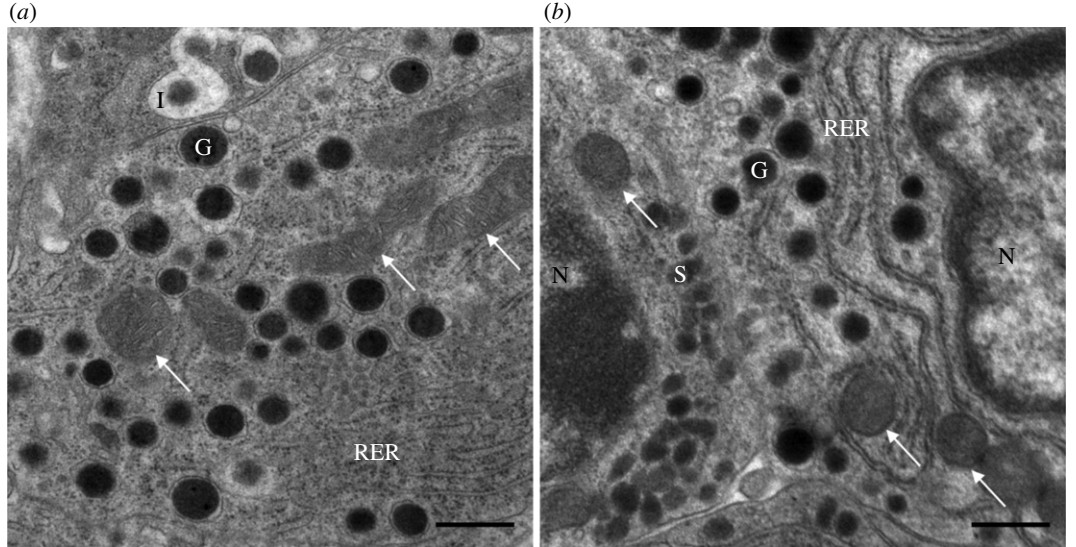

**Figure 2.** Ultrastructure of α-cells under the electron microscope. Ultrathin section of the pancreatic endocrine cells: (*a*) control, well-developed mitochondria; (*b*) WD, a lower density of mitochondria, a less expressed matrix and a decreased number of cristae. G, glucagon granule; I, insulin granule; N, nucleus; RER, rough endoplasmic reticulum; S, somatostatin granule; the white arrows point on mitochondria. Scale bar, 500 nm.

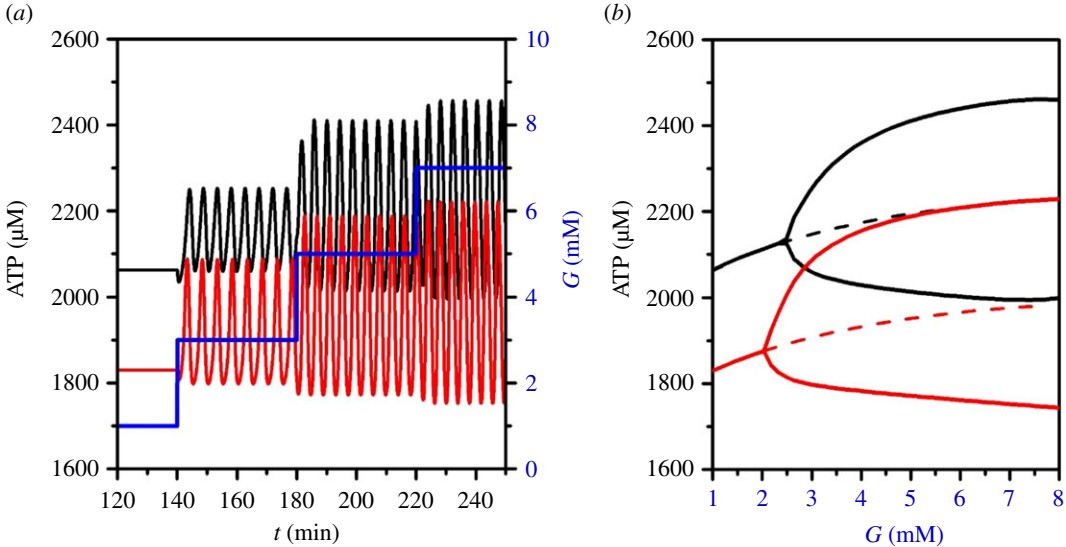

**Figure 3.** Varying ATP levels at different glucose concentrations. (*a*) Changes in ATP concentrations at different glucose concentrations (blue line) under physiological conditions (black line) and with mitochondrial oxidation ability reduced by 30% (red line); (*b*) the corresponding bifurcation diagram of ATP oscillations.

### 3.2.1. Simulating ATP production

Mitochondria are the main source of intracellular ATP. Under physiological conditions, when blood glucose levels decrease, mitochondria efficiently oxidize FFA and produce enough ATP to keep $K_{ATP}$-channel conductance low, which in turn provides the required secretion of glucagon. When the concentration of glucose is increased, a part of FFA oxidation in mitochondria is replaced by glucose. Additionally, glucose is metabolized via glycolysis in the cytoplasm. Figure 3 shows how ATP levels are increased in α-cells upon glucose stimulation with normal and reduced (30%) mitochondrial activity. The latter is being modelled as an inhibition in the glucose oxidation and β-oxidation rate, which in turn decreases the cytosolic ATP/ADP ratio (see equations (2.7)–(2.11)). With both normal and decreased function of mitochondria, the oscillatory values of ATP become larger when the glucose concentration is increasing; however, the rise in ATP is lower in the case of dysfunctional

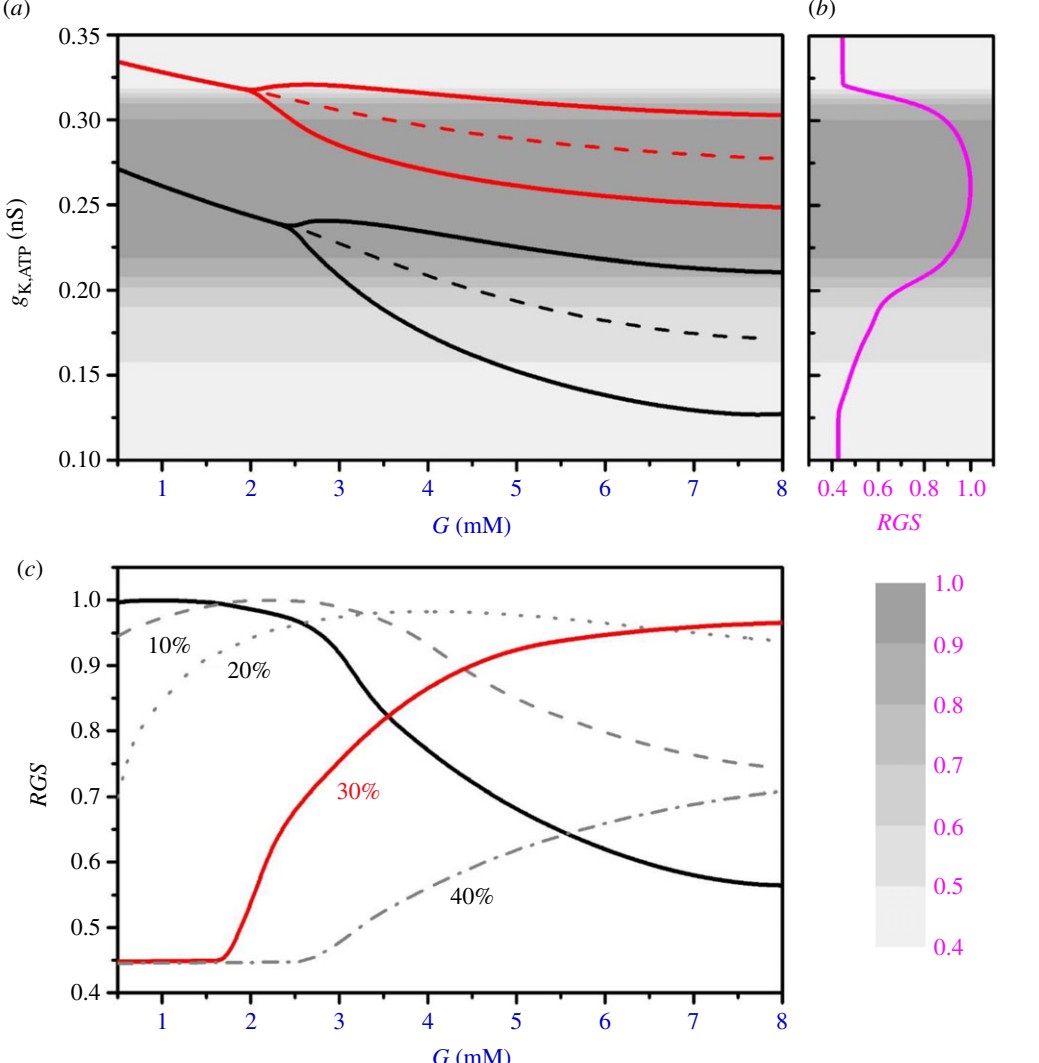

**Figure 4.** Glucose regulates K$_{ATP}$-channel conductance and glucagon secretion. (*a*) K$_{ATP}$-channel conductance, $g_{K,ATP}$, as a function of glucose concentration for normal (black line) and 30% reduced (red line) mitochondrial oxidation; (*b*) the corresponding relative glucagon secretion (*RGS*) (purple line); (*c*) *RGS* in dependence on glucose concentration for physiological conditions (black line), and for different levels of mitochondrial dysfunction, i.e. less efficient mitochondrial ATP production: 10% (dashed line), 20% (dotted line), 30% (red line) and 40% (dash-dotted line).

mitochondria, which is shown in figure 3*a*, and the corresponding bifurcation with indicated amplitudes in ATP concentrations is given in figure 3*b*.

### 3.2.2. Simulating glucagon secretion

Glucagon secretion depends mostly on the level of ATP via the conductance of K$_{ATP}$ channels, electrical activity and intracellular Ca$^{2+}$ concentration. ATP is needed to decrease the K$_{ATP}$-channel open probability; hence, the K$_{ATP}$-channel conductance, $g_{K,ATP}$, decreases with increasing ATP concentrations at higher glucose levels (figure 4*a*). When mitochondrial oxidation is impaired, the K$_{ATP}$-channel conductance shifts to higher values, accounting for dysregulated K$_{ATP}$-channel conductance (see the red line in figure 4*a*, which corresponds to mitochondrial function decreased by 30%). This in turn deteriorates glucagon secretion as well. The dependence of glucagon secretion on K$_{ATP}$-channel conductance, $g_{K,ATP}$, is presented in figure 4*b*.

Glucagon secretion for different glucose concentrations is shown in figure 4*c*. Under physiological conditions with intact mitochondria (black line in figure 4*c*), the highest glucagon secretion occurs at low glucose levels, in particular when glucose drops considerably below the physiological value. This is crucial to avoid fatal consequences of hypoglycaemia. On the other hand, our computational results

indicate an effective 'glycolytic switch' for reducing glucagon secretion when glucose rises above the physiological levels of about 4–5 mM. However, when the ATP production is impaired due to mitochondrial dysfunction, glucagon secretion is pathologically 'right-shifted' (see the red line in figure 4c, and also the dashed, dotted and dash-dotted lines for 10%, 20% and 40% of mitochondrial dysfunction, respectively).

The results of the sensitivity analysis of the model, provided in the electronic supplementary material, show that the model robustly predicts the glucagon secretion in dependence on variations in model parameters. For a broader range of parameter changes, the model predictions realistically reflect those shown here for the reference set of the model parameters (tables 1–3). Moreover, the sensitivity analysis gives a more in-depth insight into the model behaviour revealing the crucial parameters that considerably affect the energetic processes in α-cells and have the main impact on glucagon secretion, which additionally highlights the bioenergetic disruptions that might be related, or even unrelated, to the mitochondrial dysfunction presented here.

# 4. Discussion

We generated and evaluated a novel computational model for glucagon secretion in which mitochondrial efficiency can be varied to simulate glucagon secretion in dependence on different levels of mitochondrial ability to produce ATP. The basic premise of the model is that the effectiveness of mitochondrial function is altered upon a specific diet and that the mitochondrial dysfunction is linked to the pathological morphological changes. The modelling was inspired by the experimental results of our group showing that the morphology of mitochondria in α-cells is altered in mice subjected to WD that were obese and developed partially decompensated T2DM with hyperglycaemia and hyperinsulinaemia (see electronic supplementary material, table S1). The changes in these α-cells were remarkable. Electron microscopy images of ultrathin sections of pancreatic tissue indicate that mitochondria in α-cells of diabetic mice are swollen, with a dissolved matrix, and with a considerably reduced number of cristae. A direct comparison of the WD-induced mitochondrial alterations in α-cells with the control group is shown in figure 2. The differences between the mitochondrial morphology in mice subjected to WD and the control group are further supported in the electronic supplementary material. The electron microscopy images taken from all animals subjected to WD (electronic supplementary material, figures S1–S5 and S8) show characteristic alterations in mitochondrial structure, similar to that described in figure 2b; whereas the electron microscopy images for animals from the control group (electronic supplementary material, figures S9–S12) share the similar mitochondrial structures as presented in figure 2a.

Our model predicts, with a high level of robustness to model parameters (electronic supplementary material, Sensitivity analysis), that less efficient mitochondria in α-cells of T2DM mice induce glucagon dysregulation. The glucose-dependent secretion of glucagon is right-shifted, characterized by a lack of glucagon secretion at low glucose and oversecretion at high glucose concentrations. The lack of glucagon secretion at low glucose would eventually lead to hypoglycaemia, which would be in accordance with the experimental observations of Kusminski et al. [53]. Their findings suggest that an α-cell-specific induction of mitoNEET, a dimeric mitochondrial membrane protein, perturbs glucagon homeostasis and causes fasting-induced hypoglycaemia.

Furthermore, Kusminski et al. [53] show remarkable evidence of normoglycaemic retainment when altering mitochondrial function in both α-cells and β-cells. A dual overexpression of mitoNEET in both α-cells and β-cells is protective against the mitoNEET-driven β-cell dysfunction typically observed in β-cell-specific induction of mitoNEET [53]. Although our experimental set-up and the present results do not allow us to make a direct and more comprehensive comparison with the results with mitoNEET, there are some interesting observations. For example, we also found mitochondrial alterations in β-cells (electronic supplementary material, figures S6 and S7), and according to the dual mitochondrial impairment with the induction of mitoNEET [53], this might lead to more normoglycaemic conditions. We do not have any results on that, but it might be a matter of stage of the T2DM development. In some cases, we have observed that mitochondrial alterations are more pronounced in β-cells than in α-cells (e.g. electronic supplementary material, figure S5, where also autolysosomes are present). We can only hypothesize that in the course of T2DM development, the mitochondrial destruction first affects β-cells and then α-cells. Although this is only a hypothesis, a 'mild', not too destructive, α-cell perturbation at an early stage of T2DM development, would make sense in the context of the revealed protective effects of α-cell-activated mediators on the neighbouring

β-cells [53]. Additionally, the more pronounced mitochondrial alterations in β-cells than in α-cells may also be related with the overall higher resistance of α-cells to oxidative stress due to UCP2 overexpression [54] and effective protection by abundant anti-apoptotic protein expression Bcl2l1 (also known as Bcl-xL) [27]. However, again, we lack the experimental evidence on that, and further experimental studies would be needed to explain these phenomena.

There is a general lack of studies in α-cells, and much more knowledge has been accumulated about changes in β-cell morphology, their inability of insulin secretion and even their complete destruction during the course of diabetes [15,20–22,24,55]. Here, we contribute to a better understanding of the processes in α-cells by linking the experimental observation of altered mitochondrial structure in α-cells with a biophysical model which is able to account for dysregulated glucagon secretion usually observed in T2DM. The theoretical prediction of the right shift in glucagon secretion of T2DM mouse substantiates previous studies showing that T2DM is associated with the loss of glucose-induced suppression of glucagon secretion, when the physiological threshold is reached, and stimulation may occur instead [10,56]. It has been shown that this dysregulated glucagon secretion is intrinsic to the islet [10]. Indeed, it was experimentally demonstrated that the glucose-induced inhibition of $K_{ATP}$ channels is the key intrinsic mechanism in α-cells that is responsible for the inhibition of glucagon secretion. Moreover, the glucagon secretory defects associated with T2DM were mimicked by experimental conditions leading to a small increase in $K_{ATP}$-channel conductance [10]. On the other hand, it has been shown that glucose-regulated glucagon secretion can be restored in diabetic or metabolically compromised islets by low concentrations of the $K_{ATP}$-channel blocker tolbutamide [10]. Similarly, for UCP2-deleted (UCP2$^{-/-}$) mouse α-cells, the impaired glucagon secretion could be restored by slightly opening $K_{ATP}$ channels with a low dose of diazoxide (1 µmol l$^{-1}$) [57]. Because UCP2-deletion (UCP2$^{-/-}$) increases ATP levels and decreases $K_{ATP}$-channel conductance, the treatment with diazoxide, a $K_{ATP}$-channel opener, can correct the glucagon secretion defect observed in UCP2-deleted α-cells. Our mathematical model incorporates this $K_{ATP}$-channel dynamic as characterized experimentally. However, several other mechanisms, as discussed in the continuation, are also important, and should be considered in further extensions of the mathematical model presented here.

In particular, at higher glucose concentrations, paracrine effects of somatostatin on glucagon secretion are important [58,59], and under specific circumstances, as demonstrated for genetically modified mice, completely $K_{ATP}$-channel-independent mechanisms might also be involved [60,61]. Although there is less doubt that $Ca^{2+}$ is required for activation of glucagon granules, there is even more evidence that the glucagon secretion is additionally regulated by cyclic AMP (cAMP) as a second messenger [62,63]. The hypothesis is that glucose concentrations, at least in the hypoglycaemia range, can directly influence the cAMP concentrations and modulate the glucagon secretion. If $Ca^{2+}$ is a critical trigger of glucagon exocytosis in α-cells, then the magnitude of glucagon secretion appears to be mainly controlled by cAMP-mediated amplification of granule exocytosis [63]. In future, these findings need to be incorporated into a more detailed model of glucagon secretion in α-cells, possibly also together with the often neglected α-cell heterogeneity, as outlined in a recent computational study [64].

It has been recently shown that efficient energy production in α-cells, in particular via fatty acid oxidation in mitochondria, is required for normal glucagon secretion, and that inhibiting this metabolic pathway profoundly decreases glucagon output. Interestingly, this is not mediated by the $K_{ATP}$-channel, but instead due to reduced operation of the Na$^+$/K$^+$ pump [3]. These data suggest that glucagon secretion at low levels of glucose is driven by fatty acid metabolism, and that the Na$^+$/K$^+$ pump is an important ATP-dependent regulator of α-cell function. Thus, $K_{ATP}$ channels are not the only regulatory mechanism responsible for regulation/dysregulation of glucagon secretion in α-cells; in addition, the energy demanding Na$^+$/K$^+$ pumps represent another important co-regulator of glucagon secretion. When mitochondria are less efficient, providing less energy, also the Na$^+$/K$^+$ pumps are affected, and this needs to be taken into account (see Material and methods for a detailed description on how this is implemented in our mathematical model).

Mitochondrial functioning could also be directly impaired by hyperglycaemia, which is a hallmark of T2DM. Recent evidence [65] shows that hyperglycaemia might impact glucagon secretion through an increased Na$^+$ uptake. The elevation in intracellular Na$^+$ concentration leads to acidification, as a direct consequence of a lower Na$^+$ gradient across the plasma membrane that cannot drive efficiently the uphill transport of H$^+$. The cytoplasmic acidification results in a marked reduction in intramitochondrial (matrix) pH, which leads to a lower H$^+$ flux through ATP synthase and hence to an impaired ATP production. These recent findings are fully in line and further support our results indicating that the energy-driven processes, mainly provided by mitochondria, are crucial for normal regulation of glucagon secretion in α-cells.

When less energy is produced in the cell, due to mitochondrial impairment, the ATPases of the cell and ER membrane are also affected. The ER and related ER stress is another important issue that needs to be considered under the condition of mitochondrial dysfunction, also because of the importance of the mitochondria-associated ER membrane (MAM) [66,67]. In general, it is known that α-cells are much more resistant to ER stress; however, the influence of reduced energy production on the ER in α-cells is not well understood. Although it has been shown that efficient energy production via both FFA [3] and glucose [68] is indispensable for normal physiological glucagon secretion, the exact interplay between the energy-providing processes, in particular in mitochondria, and the energy consumption by the ER and plasma membrane ATPases, the ATP-driven ion exchangers and ion channels need to be further investigated.

In addition to glucose and FFA studied here, amino acids play an important role in glucagon secretion. There is a large body of evidence that glucagon regulates amino acid metabolism at a systemic level, probably even more efficiently than the glucose homeostasis [69,70]. By way of a feedback loop, an elevation in circulating amino acids causes enormous glucagon secretion, known for decades [71], e.g. an intravenous arginine infusion of 5 g may result in a 10-fold increase in plasma glucagon level [72]. Amino acids also promote α-cell proliferation via a nutrient-sensing circuit [73,74]. Whereas the signalling role of amino acids on glucagon secretion is well established, its contribution to the energy production in α-cells is of much less importance. In particular, in hypoglycaemia, it is hardly to expect that amino acids would be used for ATP production in α-cells, especially in significant quantities. The glucagon-induced skeletal muscle wasting aims to supply amino acids as a gluconeogenic precursor. Importantly, amino acids do not fuel ATP production in hepatocytes, but instead, the hepatic FFA oxidation is enhanced to supply the energy required to sustain gluconeogenesis [75]. A similar scenario, with a preference for FFA consumption in hypoglycaemia, would be expected in α-cells. If under specific conditions there were some contribution of amino acids to the energy production in α-cells, from the perspective of our mathematical model, it only requires a separate quantification of this process; however, no qualitative changes in the model predictions would be expected.

Why and how exactly the mitochondria in α-cells are damaged in the course of diabetes development remains a matter of further studies. Our first evidence of structural changes in the shape of mitochondria and alterations in their inner structure needs to be further evaluated, first, on larger samples of α-cells, and second, by providing quantitative analyses of different morphological parameters concerning the main characteristics of mitochondria in α-cells. Moreover, the link between the structural changes and the functioning of mitochondria needs to be established. Further experiments with, for example, tolbutamide or diazoxide would be needed to evaluate the mitochondrial function, or even more sophisticated methods for changes in mitochondrial DNA, and some metabolomics, would be needed to see the physiological processes being altered in the process of mitochondrial dysfunction.

Further studies will also be needed to investigate the complex interplay between the energy-driven, anabolic and signalling mechanisms. Recognizing the mitochondria as multi-functional bioenergetic, biosynthetic and signalling organelles [76–78] may result in completely new clinical treatments of diabetes. We may in the future improve the future treatment of diabetes considerably by stimulating and regenerating the mitochondrial function, partly already with increased physical activity and weight loss that restore mitochondrial content and functional capacity, particularly in skeletal muscle [79], and with the development of new medications stimulating mitogenesis and influencing mitochondrial function more efficiently than the currently known agents, e.g. coenzyme Q10 [15,80] or metformin [81–83].

Ethics. The study was conducted in strict accordance with all national and European recommendations pertaining to care and work with experimental animals, and all efforts were made to minimize suffering of animals. The protocol was approved by the Veterinary Administration of the Republic of Slovenia (permit no. U34401-12/2015/3). Animal care and experimentations in Sweden were approved by the Animal Experiment Ethics Committee at Karolinska Institutet in compliance with other statutes and regulations relating to animals and experiments in Stockholm, Sweden (permit no. N144/14).

Data accessibility. The source code for the computational model is available from the Dryad Digital Repository: https://doi.org/10.5061/dryad.9n2k1vk [84].

Authors' contributions. M.M., V.G. and R.M. designed the study. V.G., R.M., S.L., M.G., J.D., A.S., M.P. and M.M. wrote the paper. V.G. and R.M. performed the numerical simulations and the analyses. I.V.-A., P.-O.B., J.D. and A.S. took care of the mice colony and performed the whole-animal measurements and systemic tests. I.V.-A. and P.-O.B. also edited the manuscript. Tissue preparation and transmission electron microscopy was performed by S.L and G.L.

Competing interests. M.P. was a member of the Royal Society Open Science Editorial Board at the time of submission and publication of this paper, but was not involved in the assessment of the manuscript.

Funding. This work was supported by the Slovenian Research Agency (research core funding, nos. P1-0055, P1-0403 and P3-0396, as well as research projects, nos. J3-9289, N3-0048, I0-0029, J1-7009, J7-7226, J4-9302, J1-9112 and P1-0403).

This work was also supported by the Swedish Diabetes Association, Funds of Karolinska Institutet, The Swedish Research Council, Novo Nordisk Foundation, The Family Erling-Persson Foundation, Strategic Research Program in Diabetes at Karolinska Institutet, The Family Knut and Alice Wallenberg Foundation, The Stichting af Jochnick Foundation, Skandia Insurance Company, Ltd, Diabetes and Wellness Foundation, The Bert von Kantzow Foundation, Svenska Diabetesstiftelsen, AstraZeneca, Swedish Association for Diabetology and The ERC-2013-AdG 338936-BetaImage. We thank Mark Graham, now retired from IONIS, for support. The funders had no role in study design, data collection and analysis, decision to publish or preparation of the manuscript.

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
