## [Reviewer comments · Royal Society Open Science]

Review History

RSOS-191171.R0 (Original submission)

Review form: Reviewer 1

Is the manuscript scientifically sound in its present form?

Yes

Are the interpretations and conclusions justified by the results?

Yes

Is the language acceptable?

Yes

Do you have any ethical concerns with this paper?

No

Have you any concerns about statistical analyses in this paper?

No

Recommendation?

Accept with minor revision (please list in comments)

Comments to the Author(s)

In their work entitled "Modeling dysregulated glucagon secretion in type 2 diabetes based on experimentally observed mitochondrial defects in pancreatic alpha cells of mice" the authors review recent experimental evidence on the role played by dysregulated glucagon in diabetes emergence and construct a novel computational model that links alpha-cell metabolism with their electrical activity and glucagon secretion. The model and its parameters are suitably adjusted in order to fit several aspects of model predictions with experimental findings. This work suggests that the attenuated mitochondrial oxidative capacity could explain the dysregulation of glucagon secretion that is typical of type 2 diabetes.

I find that the topic considered in this manuscript is timely and of great interest for a broad community of scientists; the underlying analysis is sound and the length appropriate. I therefore believe that it is worth of publication on Royal Society Open Science.

I have only a point that I would like the authors to address, at least succinctly.

The model introduced in this work displays overall a large number of parameters (Tables 1, 2, 3 all together contain roughly a hundred of parameters). Could you possibly discuss the robustness of your model with respect to these parameters? Otherwise stated, how sensitive is the outcome of the model with respect these parameters? Are these parameters all empirically measurable? Could you comment on their value plausibility?

Finally, please, also notice a couple of typos
 - page 2, line 43: "microcopy" -> "microscopy"
 - page 4, line 17: "were" is repeated twice

Review form: Reviewer 2

Is the manuscript scientifically sound in its present form?

No

Are the interpretations and conclusions justified by the results?

Yes

Is the language acceptable?

Yes

Do you have any ethical concerns with this paper?

Yes

Have you any concerns about statistical analyses in this paper?

No

Recommendation?

Major revision is needed (please make suggestions in comments)

Comments to the Author(s)

The paper by Grubelnik and colleagues addresses a potentially important aspect of type 2 diabetes, namely alpha-cell mitochondrial defects. The taking into account of free fatty acids as

stimulants for glucagon secretion is a strength of the present study. Alpha-cell mitochondria of mice fed a high fat diet were structurally altered which provides the rationale for building the mathematical model. The mathematical model is not however, to my understanding, based on these experimental findings but rather on previous work on glucagon secretion by other groups.

My comments are as follows:

1. The experimental procedures regarding the experimental animals is not sufficiently outlined. Please fill out and attach the ARRIVE guidelines checklist (<https://royalsociety.org/journals/ethics-policies/research-ethics/>).

For example:

-It is not clear how many animals in total were used. It is mentioned that tissue of 6 mice of each group were used for the electron microscopy but it is not clear if that were all animals used from the beginning of the experiment.

-It is not stated whether animals of both sexes were used, to the best of my knowledge it is only male C57BL/6 mice that develop hyperglycemia upon high-fat feeding.

-Are the controls littermate controls?

-Were the diets isocaloric but of different composition or was the western diet higher in energy content?

-The body weight development of the mice is not provided.

-How were the animals sacrificed?

2. In interpreting the results, the western diet-fed mice are referred to as 'type 2 diabetic mice'. There is no data on this. Please provide data to show the development of hyperglycemia over time.

3. Were any steps taken to reduce subjective bias? For example was the person evaluating the electron microscopy slides blinded to the treatment group?

4. I lack a summary and conclusion at the end of the discussion. Future topics to study is discussed in length but I lack the authors' take on the actual conclusion and what is the most important finding of the study.

4. The rationale of the model is based on the experimental finding of altered mitochondrial structure upon high-fat diet feeding.

5. To my understanding the results of the animal experiments is not used as the base for the mathematical model which seems based on previous results from other groups. To me the title seems somewhat misleading. Please rephrase the title to not give the impression that the models is based on the experiments in the paper. Or if I am mistaken, please clarify how your experimental findings are incorporated in the mathematical model.

6. The structural alterations of the mitochondria is illustrated by one representative figure. Please provide a representative figure from each animal as part of the supplementary material.

7. The experiments of the paper show that the structure of mitochondria are altered with high-fat diet but the function is not studied. In previous work by Kusminski and colleagues (<https://www.ncbi.nlm.nih.gov/pmc/articles/PMC5310214/>) where the function of alpha-cell mitochondria are experimentally disrupted the outcome was not an alteration in glucagon secretion in the way suggested by the model of the present study. Please consider and discuss the discrepancies in the findings of the present study and the previous experimental findings by Kusminski and colleagues.

8. The model includes input from glucose and free fatty acids, which is good. However, it is increasingly clear that amino acids are of great importance in the regulation of glucagon

secretion. Please discuss the rationale of considering glucose and free fatty acids but not amino acids when building the model.

9. Were the structure of the beta-cell mitochondria altered at the same time? Is it possible for the authors to tell whether structural alterations of mitochondria happens in parallel in alpha and beta cells or if one precedes the other?

10. Please include in the paper the rationale for the use of this particular animal model to study the changes in mitochondria in type 2 diabetes.

Decision letter (RSOS-191171.R0)

24-Sep-2019

Dear Professor Perc,

The editors assigned to your paper ("Modeling dysregulated glucagon secretion in type 2 diabetes based on experimentally observed mitochondrial defects in pancreatic alpha cells of mice") have now received comments from reviewers. We would like you to revise your paper in accordance with the referee and Associate Editor suggestions which can be found below (not including confidential reports to the Editor). Please note this decision does not guarantee eventual acceptance.

Please submit a copy of your revised paper before 17-Oct-2019. Please note that the revision deadline will expire at 00.00am on this date. If we do not hear from you within this time then it will be assumed that the paper has been withdrawn. In exceptional circumstances, extensions may be possible if agreed with the Editorial Office in advance. We do not allow multiple rounds of revision so we urge you to make every effort to fully address all of the comments at this stage. If deemed necessary by the Editors, your manuscript will be sent back to one or more of the original reviewers for assessment. If the original reviewers are not available, we may invite new reviewers.

- Data accessibility

If you wish to submit your supporting data or code to Dryad (<http://datadryad.org/>), or modify your current submission to dryad, please use the following link:
<http://datadryad.org/submit?journalID=RSOS&manu=RSOS-191171>

- Competing interests

- Authors' contributions

- Acknowledgements

- Funding statement

Kind regards,

Andrew Dunn

on behalf of Professor Anotida Madzvamuse (Associate Editor) and Pietro Cicuta (Subject Editor)
openscience@royalsociety.org

Associate Editor's comments (Professor Anotida Madzvamuse):

Associate Editor: 1

Comments to the Author:

Given the major comments raised by both reviewers, major revision of the manuscript is required. In particular, authors must address the merits of the model and what data was used for its formulation as well as carrying out detailed sensitivity analysis of the parameters. Finally, detailed and appropriate conclusions must be provided. Acceptance of the revised manuscript will hugely depend on how these are addressed and authors must provide a point-by-point rebuttal/explanation of how their responses to reviewers' comments.

Comments to Author:

Reviewers' Comments to Author:

Reviewer: 1

Comments to the Author(s)

In their work entitled "Modeling dysregulated glucagon secretion in type 2 diabetes based on experimentally observed mitochondrial defects in pancreatic alpha cells of mice" the authors review recent experimental evidence on the role played by dysregulated glucagon in diabetes emergence and construct a novel computational model that links alpha-cell metabolism with their electrical activity and glucagon secretion. The model and its parameters are suitably adjusted in order to fit several aspects of model predictions with experimental findings. This work suggests that the attenuated mitochondrial oxidative capacity could explain the dysregulation of glucagon secretion that is typical of type 2 diabetes.

I find that the topic considered in this manuscript is timely and of great interest for a broad community of scientists; the underlying analysis is sound and the length appropriate. I therefore believe that it is worth of publication on Royal Society Open Science.

I have only a point that I would like the authors to address, at least succinctly.

The model introduced in this work displays overall a large number of parameters (Tables 1, 2, 3 all together contain roughly a hundred of parameters). Could you possibly discuss the robustness of your model with respect to these parameters? Otherwise stated, how sensitive is the outcome of the model with respect these parameters? Are these parameters all empirically measurable? Could you comment on their value plausibility?

Finally, please, also notice a couple of typos

- page 2, line 43: "microcopy" -> "microscopy"

- page 4, line 17: "were" is repeated twice

Reviewer: 2

Comments to the Author(s)

The paper by Grubelnik and colleagues addresses a potentially important aspect of type 2 diabetes, namely alpha-cell mitochondrial defects. The taking into account of free fatty acids as stimulants for glucagon secretion is a strength of the present study. Alpha-cell mitochondria of mice fed a high fat diet were structurally altered which provides the rationale for building the mathematical model. The mathematical model is not however, to my understanding, based on these experimental findings but rather on previous work on glucagon secretion by other groups.

My comments are as follows:

1. The experimental procedures regarding the experimental animals is not sufficiently outlined.

Please fill out and attach the ARRIVE guidelines checklist (<https://royalsociety.org/journals/ethics-policies/research-ethics/>).

For example:

-It is not clear how many animals in total were used. It is mentioned that tissue of 6 mice of each group were used for the electron microscopy but it is not clear if that were all animals used from the beginning of the experiment.

-It is not stated whether animals of both sexes were used, to the best of my knowledge it is only male C57BL/6 mice that develop hyperglycemia upon high-fat feeding.

-Are the controls littermate controls?

-Were the diets isocaloric but of different composition or was the western diet higher in energy content?

-The body weight development of the mice is not provided.

-How were the animals sacrificed?

2. In interpreting the results, the western diet-fed mice are referred to as 'type 2 diabetic mice'. There is no data on this. Please provide data to show the development of hyperglycemia over time.

3. Were any steps taken to reduce subjective bias? For example was the person evaluating the electron microscopy slides blinded to the treatment group?

4. I lack a summary and conclusion at the end of the discussion. Future topics to study is discussed in length but I lack the authors' take on the actual conclusion and what is the most important finding of the study.

4. The rationale of the model is based on the experimental finding of altered mitochondrial structure upon high-fat diet feeding.

5. To my understanding the results of the animal experiments is not used as the base for the mathematical model which seems based on previous results from other groups. To me the title seems somewhat misleading. Please rephrase the title to not give the impression that the models is based on the experiments in the paper. Or if I am mistaken, please clarify how your experimental findings are incorporated in the mathematical model.

6. The structural alterations of the mitochondria is illustrated by one representative figure. Please provide a representative figure from each animal as part of the supplementary material.

7. The experiments of the paper show that the structure of mitochondria are altered with high-fat diet but the function is not studied. In previous work by Kusminski and colleagues (<https://www.ncbi.nlm.nih.gov/pmc/articles/PMC5310214/>) where the function of alpha-cell mitochondria are experimentally disrupted the outcome was not an alteration in glucagon secretion in the way suggested by the model of the present study. Please consider and discuss the discrepancies in the findings of the present study and the previous experimental findings by Kusminski and colleagues.

8. The model includes input from glucose and free fatty acids, which is good. However, it is increasingly clear that amino acids are of great importance in the regulation of glucagon secretion. Please discuss the rationale of considering glucose and free fatty acids but not amino acids when building the model.

9. Were the structure of the beta-cell mitochondria altered at the same time? Is it possible for the authors to tell whether structural alterations of mitochondria happens in parallel in alpha and beta cells or if one precedes the other?

10. Please include in the paper the rationale for the use of this particular animal model to study the changes in mitochondria in type 2 diabetes.

Author's Response to Decision Letter for (RSOS-191171.R0)

See Appendix A.

RSOS-191171.R1 (Revision)

Review form: Reviewer 2

Is the manuscript scientifically sound in its present form?

Yes

Are the interpretations and conclusions justified by the results?

Yes

Is the language acceptable?

Yes

Do you have any ethical concerns with this paper?

Yes

Have you any concerns about statistical analyses in this paper?

No

Recommendation?

Accept as is

Comments to the Author(s)

I'm quite happy with the responses from the authors and the changes made to the manuscript.

Decision letter (RSOS-191171.R1)

16-Dec-2019

Dear Professor Perc,

It is a pleasure to accept your manuscript entitled "Modelling of dysregulated glucagon secretion in type 2 diabetes by considering mitochondrial alterations in pancreatic alpha cells" in its current form for publication in Royal Society Open Science. The comments of the reviewer(s) who reviewed your manuscript are included at the foot of this letter.

on behalf of Professor Anotida Madzvamuse (Associate Editor) and Pietro Cicuta (Subject Editor)
openscience@royalsociety.org

Associate Editor Comments to Author (Professor Anotida Madzvamuse):

First Comments to the Author:

Thanks for addressing comprehensively the reviewers' comments. I would like to get one final review to make sure that the new draft meets all the comments raised.

Final Comments to the Author:

Accept the manuscript

Reviewer comments to Author:

Reviewer: 2

Comments to the Author(s)

I'm quite happy with the responses from the authors and the changes made to the manuscript.

Appendix A

RESPONSE TO REVIEWER'S COMMENTS

Reviewer: 1

In their work entitled “Modeling dysregulated glucagon secretion in type 2 diabetes based on experimentally observed mitochondrial defects in pancreatic alpha cells of mice” the authors review recent experimental evidence on the role played by dysregulated glucagon in diabetes emergence and construct a novel computational model that links alpha-cell metabolism with their electrical activity and glucagon secretion. The model and its parameters are suitably adjusted in order to fit several aspects of model predictions with experimental findings. This work suggests that the attenuated mitochondrial oxidative capacity could explain the dysregulation of glucagon secretion that is typical of type 2 diabetes.

I find that the topic considered in this manuscript is timely and of great interest for a broad community of scientists; the underlying analysis is sound and the length appropriate. I therefore believe that it is worth of publication on Royal Society Open Science.

Comment

I have only a point that I would like the authors to address, at least succinctly. The model introduced in this work displays overall a large number of parameters (Tables 1, 2, 3 all together contain roughly a hundred of parameters). Could you possibly discuss the robustness of your model with respect to these parameters? Otherwise stated, how sensitive is the outcome of the model with respect these parameters? Are these parameters all empirically measurable? Could you comment on their value plausibility?

Response

We completely agree with the reviewer that the robustness of the model with respect to the parameters is a crucial issue. Therefore, we have conducted a comprehensive sensitivity analyses of the model. The sensitivity of the model predictions has been carried out for all parameters that have been involved in linking the parts of previous published models, and have been used as plug-in elements in our model, as well as for all the parameters that have been newly involved in our model. The main results of this analysis are presented in the main text, and the complete results of the analysis, with all graphical presentations, are included in the Supplement.

Comment

Finally, please, also notice a couple of typos

- page 2, line 43: “microcopy” -> “microscopy”
- page 4, line 17: “were” is repeated twice

Response

Thanks for making us aware of the typos; we have corrected them.

Reviewer: 2

The paper by Grubelnik and colleagues addresses a potentially important aspect of type 2 diabetes, namely alpha-cell mitochondrial defects. The taking into account of free fatty acids as stimulants for glucagon secretion is a strength of the present study. Alpha-cell mitochondria of mice fed a high fat diet were structurally altered which provides the rationale for building the mathematical model. The mathematical model is not however, to my understanding, based on these experimental findings but rather on previous work on glucagon secretion by other groups.

Comment

1. The experimental procedures regarding the experimental animals is not sufficiently outlined. Please fill out and attach the ARRIVE guidelines checklist (<https://royalsociety.org/journals/ethics-policies/research-ethics/>).

For example:

-It is not clear how many animals in total were used. It is mentioned that tissue of 6 mice of each group were used for the electron microscopy but it is not clear if that were all animals used from the beginning of the experiment.

-It is not stated whether animals of both sexes were used, to the best of my knowledge it is only male C57BL/6 mice that develop hyperglycemia upon high-fat feeding.

-Are the controls littermate controls?

-Were the diets isocaloric but of different composition or was the western diet higher in energy content?

-The body weight development of the mice is not provided.

-How were the animals sacrificed?

Response:

According to the reviewer's comments we have completely rewritten the experimental part of the section Materials and Methods. Based on the suggested AARIVE Guidelines, in the revised version of the manuscript all the required elements of the experimental procedures are provided. In Section 3.1. data about the animals, their number, sex, body weight, diet, housing, etc. have been included. The Section 3.1. contains much broader spectrum of data; therefore, it has been renamed as "Animals, diets, and metabolic parameters".

Comment

2. In interpreting the results, the western diet-fed mice are referred to as 'type 2 diabetic mice'. There is no data on this. Please provide data to show the development of hyperglycemia over time.

Response:

We agree with the reviewer that referring the WD-fed mice to be diabetic requires additional experimental data. We have included the data in the Supplement, and we made a reference to the Supplement in the beginning of the section Results. The measurements were provided by the laboratory in Karolinska Institute, Sweden. Therefore, we have included two additional coauthors Professor Per-Olof Berggren and Professor Ismael Valladolid-Acebes from that institution. We also added an additional reference to their previous publication showing many characteristics of this model (Valladolid-Acebes et al., 2015).

Comment

3. Were any steps taken to reduce subjective bias? For example was the person evaluating the electron microscopy slides blinded to the treatment group?

Response:

All the experiments concerning the electron microscopy have been conducted and evaluated blinded to the treatment groups. This is emphasized in the body text, in the Section 3.2., where we present the preparation and the analysis of the tissue.

Comment

4. I lack a summary and conclusion at the end of the discussion. Future topics to study is discussed in length but I lack the authors' take on the actual conclusion and what is the most important finding of the study.

Response:

We agree with the reviewer that a summarized presentation of the most important findings of the study was missing. We even feel that this summary is so important that it has to be emphasized in the beginning of the Discussion, immediately after the presentation of the results (in the previous section Results). Therefore, according to the reviewer's suggestion, we have completely rewritten the first paragraph of the Discussion, where now the most important findings of our study are summarized and further discussed.

Comment

4. The rationale of the model is based on the experimental finding of altered mitochondrial structure upon high-fat diet feeding.

5. To my understanding the results of the animal experiments is not used as the base for the mathematical model which seems based on previous results from other groups. To me the title seems somewhat misleading. Please rephrase the title to not give the impression that the models is based on the experiments in the paper. Or if I am mistaken, please clarify how your experimental findings are incorporated in the mathematical model.

Response:

The reviewer is completely right. No quantitative data from our experiments are incorporated in the model equations. Our experiments qualitatively reveal the structural alterations in alpha cells, which gave us the inspiration for mathematical modelling of glucagon secretion in dependence on mitochondrial dysfunction. The mathematical model is constructed on the bases of previous experimental data from other groups, as also appropriately cited throughout the manuscript. Therefore, we completely agree with the reviewer's observation and according to the reviewer's suggestion, we have changed the title of the manuscript. The title no longer contains the expression "based on experimentally revealed..."

Comment

6. The structural alterations of the mitochondria is illustrated by one representative figure. Please provide a representative figure from each animal as part of the supplementary material.

Response:

As indicated in the Response to Comments 4 and 5, our experimental observations have been basically used as the key inspiration for the mathematical modelling. Therefore, it was not our primary intent to emphasize and evaluate the experimental data to a larger extent. However, we completely agree with the referee that one figure is not enough to make a solid base for this important experimental observation, revealing the structural alterations of the mitochondria in WD-fed mice. Therefore, as suggested by the reviewer, in the Supplement we provided 12 additional images from the animals that we have investigated.

Comment

7. The experiments of the paper show that the structure of mitochondria are altered with high-fat diet but the function is not studied. In previous work by Kusminski and colleagues (<https://www.ncbi.nlm.nih.gov/pmc/articles/PMC5310214/>) where the function of alpha-cell mitochondria are experimentally disrupted the outcome was not an alteration in glucagon secretion in the way suggested by the model of the present study. Please consider and discuss the discrepancies in the findings of the present study and the previous experimental findings by Kusminski and colleagues.

Response:

We appreciate this comment very much. Thanks for making us aware about this excellent work of Kusminski et al. (2016). This study is indeed highly appropriate for a deeper discussion of our results. In the Discussion of the revised manuscript, we have added two additional paragraphs devoted to the comparison of the results provided by Kusminski et al. (2016) and our results. In the first paragraph, we point out the agreement of our model predictions concerning the right shift of glucagon secretion in diabetic mice. We emphasise in the text that the lack of glucagon secretion at low glucose would eventually lead to hypoglycemia, which would be in accordance with the experimental observations of Kusminski et al. (2016). Namely, their findings suggest that an α -cell-specific induction of mitoNEET, a dimeric mitochondrial membrane protein, perturbs glucagon homeostasis and causes fasting-induced hypoglycemia.

Furthermore, we discuss the remarkable evidence provided by Kusminski et al. (2016) that normoglycemic retention is established after altering mitochondrial function in both α -cells and β -cells. They showed that dual overexpression of mitoNEET in both α -cells and β -cells is protective against the mitoNEET-driven β -cell dysfunction typically observed in β -cell-specific induction of mitoNEET. We emphasise in the Discussion that although our experimental setup and the present results do not allow to make a direct and more comprehensive comparison to the results with mitoNEET, there are some interesting observations. For example, we also found mitochondrial alterations in β -cells (Figs. S6, S7), and according to the dual mitochondrial impairment with the induction of mitoNEET (Kusminski et al., 2016), this might lead to more normoglycemic conditions. We clearly state in the Discussion that we do not have any results on that, but we hypothesise that it might be a matter of stage of the T2DM development. Namely, in some cases, we have observed that mitochondrial alterations are more pronounced in β -cells than in α -cells (e.g. Fig. S5, where also autolysosomes are present). Therefore, we can only hypothesize that in the course of T2DM development, the mitochondrial destruction first affects β -cells and then α -cells. Although this is only a hypothesis, a “mild”, not too destructive, α -cell perturbation at an early stage of T2DM

development, would make sense in the context of the revealed protective effects of α -cell-activated mediators on the neighbouring β -cells (Kusminski et al., 2016). We further discuss that, additionally, the more pronounced mitochondrial alteration in β -cells than in α -cells may also be related with the overall higher resistance of α -cells to oxidative stress due to UCP2 overexpression (Diao et al., 2008) and effective protection by abundant anti-apoptotic protein expression Bcl2l1 (also known as Bcl-xL) (Marroqui et al., 2015). However, again, we lack the experimental evidence on that, and further experimental studies would be needed to explain these phenomena.

Comment

8. The model includes input from glucose and free fatty acids, which is good. However, it is increasingly clear that amino acids are of great importance in the regulation of glucagon secretion. Please discuss the rationale of considering glucose and free fatty acids but not amino acids when building the model.

Response:

We agree with the reviewer's comment and we have introduced a completely new paragraph in the Discussion, where we discuss the important role of amino acids. We make a clear statement that in addition to glucose and FFA, amino acids play an important role in glucagon secretion. We emphasise that there is a large body of evidence that glucagon regulates amino acid metabolism at a systemic level, probably even more efficiently than the glucose homeostasis (Unger et al., 1969; Orskov et al., 1991; Solloway et al., 2015; Dean et al., 2017; Holst et al., 2017; Janah et al., 2019). We also note that whereas the signalling role of amino acids on glucagon secretion is well established, its contribution to the energy-production in α -cells is of much less importance. In particular in hypoglycemia, it is hardly to expect that amino acids would be used for ATP production in α -cells, especially in significant quantities. The glucagon-induced skeletal muscle wasting aims to supply amino acids as a gluconeogenic precursor, and importantly, the amino acids also do not fuel ATP production in hepatocytes, but instead, the hepatic FFA oxidation is enhanced to supply the energy required to sustain gluconeogenesis (Adeva-Andany et al., 2019). A similar scenario, with a preference for FFA consumption in hypoglycemia, would also be expected in α -cells. If under specific conditions there were some contribution of amino acids to the energy production in α -cells, from the perspective of our mathematical model, it only requires a separate quantification of this process, however, no qualitative changes in the model predictions would be expected.

Comment

9. Were the structure of the beta-cell mitochondria altered at the same time? Is it possible for the authors to tell whether structural alterations of mitochondria happens in parallel in alpha and beta cells or if one preceds the other?

Response:

This is a highly interesting question, and it probably deserves a separate report on these results when available. At this stage we do not have enough experimental data to make any solid conclusions. Nevertheless, the preliminary results that we have concerning these questions are now included in the revised manuscript, as discussed in the Response to Comment 7.

Comment

10. Please include in the paper the rationale for the use of this particular animal model to study the changes in mitochondria in type 2 diabetes.

Response:

There is a plethora of diet-induced models that are utilized to study pathophysiology of T2DB. However, there are many drawbacks of the models used. Most studies use HFD to induce increase in body weight and concomitant metabolic changes that resemble T2DB seen in humans. Although effective in inducing T2DB in mice, the HFD is in its composition far from the human diet consumed in modern era, namely its composition is too high in lipids and too low in especially simple sugars. Therefore, the WD is more comparable diet intervention to resemble condition in humans. Secondly, most of the studies would start diet intervention at early age (e.g. mostly at 4 weeks of age) when the plasticity of endocrine cells is highest. Again, this strongly disagrees with the conditions in humans where typically T2DB occurs in later stages of life. Therefore, we used a model where we start diet intervention at 12 weeks of age offering animal WD. We are in process of preparing a separate paper that will describe the model in great details, ranging from subcellular morphological changes, physiological changes in endocrine cells to *in vivo* data. Since this manuscript is focused on the modelling part and the mitochondrial changes in this model served only as an inspiration for developing the modelling part, we believe that the in depth description of the animal model is beyond the scope of this paper. In any case, we ask reviewers for discretion regarding the development and benefits of the model described in this paper. Anyway, in accordance with the reviewer's suggestions, we have introduced in the text all the details about the diet in Section 3.1. "Animals, diets, and metabolic parameters", as well as provided the supplementary table (Table S1) with the hard data evidencing the glucometabolic phenotype of these mice. Finally, we added an additional reference (Valladolid-Acebes et al., 2015) to guide the reader to many more details about this particular dietary model.